# Consumer Autonomy in Generative AI Services: The Role of Task Difficulty and AI Design Elements in Enhancing Trust, Satisfaction, and Usage Intention

**DOI:** 10.3390/bs15040534

**Published:** 2025-04-15

**Authors:** Jihyung Han, Daekyun Ko

**Affiliations:** 1Research Institute of Human Ecology, Seoul National University, Seoul 08826, Republic of Korea; ellyhan@snu.ac.kr; 2Department of Consumer Science, Chungnam National University, Daejeon 34134, Republic of Korea

**Keywords:** consumer autonomy, generative AI services, trust, satisfaction, usage intention

## Abstract

As generative AI services become increasingly integrated into consumer decision making, concerns have grown regarding their influence on consumer autonomy—the extent to which individuals retain independent control over AI-assisted decisions. Although these services offer efficiency and convenience, they can simultaneously constrain consumer decision making, potentially impacting trust, satisfaction, and usage intention. This study investigates the role of perceived consumer autonomy in shaping consumer responses, specifically examining how task difficulty (Study 1) and AI service design elements—explainability, feedback, and shared responsibility (Study 2)—influence autonomy perceptions and subsequent consumer outcomes. Using two scenario-based experiments involving a total of 708 participants, the results reveal that perceived autonomy significantly enhances consumer trust, particularly in contexts involving high task difficulty. Among the tested AI design interventions, shared responsibility emerged as most effective in enhancing perceived autonomy, trust, satisfaction, and long-term engagement, whereas explainability and feedback alone showed limited impact. These findings underscore the importance of designing AI services that actively support consumer agency through user-involved decision-making frameworks rather than relying solely on passive informational transparency. Theoretical implications for consumer autonomy in AI interactions are discussed, along with practical recommendations for designing consumer-centered AI services.

## 1. Introduction

Generative AI services are fundamentally transforming consumer interactions with technology ([2]), reshaping how individuals make decisions in AI-assisted environments. The rapid adoption of tools like ChatGPT highlights the significant influence of AI-based decision support systems, providing consumers with unprecedented convenience, efficiency, and cognitive relief ([32]). However, alongside these benefits, critical concerns have emerged regarding consumer autonomy—defined as the extent to which individuals perceive themselves as active decision makers when interacting with AI systems ([48]).

Consumer autonomy is particularly crucial in AI-assisted interactions because it directly shapes consumer trust, satisfaction, and willingness to continuously engage with AI services ([12]; [24]). Specifically, an optimal balance of autonomy enhances user acceptance, whereas both excessive automation and insufficient autonomy can negatively affect consumer perceptions ([12]; [15]). Increasingly sophisticated AI-generated recommendations, though beneficial in reducing cognitive effort, can inadvertently constrain consumer independence by shifting decision authority from consumers to AI systems, ultimately undermining consumer engagement ([7]; [39]; [50]).

Recent AI ethics guidelines emphasize consumer-centered design principles that preserve user control and agency within automated environments ([15]). However, automation bias and information overload remain potential unintended consequences of AI reliance, potentially diminishing rather than enhancing perceived autonomy ([20]; [23]). While prior research extensively discusses autonomy as a stable psychological trait or external constraint, relatively little is known about how specific AI service design elements might actively shape consumers’ perceptions of autonomy and thereby influence their responses ([37]).

To address these gaps, this research empirically investigates how perceived consumer autonomy influences trust, satisfaction, and usage intention in AI-assisted decision-making contexts. Specifically, this study explores how task difficulty moderates autonomy’s impact on consumer responses and examines the effectiveness of targeted AI design elements—such as explainability, feedback, and shared responsibility—in actively enhancing consumer autonomy.

This research contributes to the growing discourse on consumer autonomy in AI interactions in several meaningful ways. First, it advances theoretical understanding by examining autonomy not merely as a passive psychological factor but as a dynamic perception actively shaped by AI design. Second, it empirically distinguishes which AI design features most effectively support consumer autonomy, providing practical guidance for developers and policymakers. Lastly, it clarifies the contexts in which autonomy-supportive features have the most substantial impact, thereby informing targeted strategies to enhance consumer trust and sustained engagement with AI services.

The following sections detail two experimental studies designed to examine these dynamics. Study 1 investigates how task difficulty moderates the relationship between perceived autonomy and consumer responses to AI services. Study 2 explores whether specific AI design elements can effectively strengthen perceptions of autonomy, subsequently influencing trust, satisfaction, and usage intentions. Together, these studies provide a comprehensive examination of how consumer autonomy can be optimally supported within AI-mediated decision-making contexts, ultimately promoting ethical, transparent, and consumer-centric AI services.

## 2. Literature Review and Hypotheses

### 2.1. Perceived Consumer Autonomy in AI Services

Consumer autonomy is a fundamental concept in decision-making research, encompassing individuals’ ability to make choices free from external constraints. Despite its significance, no universally agreed upon definition of autonomy exists ([24]). [48] ([48]) define autonomy as the ability to make and enact decisions independently, while [27] ([27]) emphasizes autonomy as decision making driven by internal motivations that reflect personal preferences. Both perspectives underscore autonomy’s centrality in consumer behavior and decision-making processes.

In AI-mediated interactions, autonomy refers either to the system’s independence or to the user’s perceived ability to make independent choices ([16]). This study focuses on the latter—the autonomy consumers perceive when interacting with AI services. Specifically, perceived autonomy is shaped by how much control and influence consumers feel over decision-making outcomes ([30]). While AI can provide helpful support, there are concerns that it may reinforce the perception that “the computer knows best”, which in turn may diminish consumers’ sense of agency ([22]; [31]). [45] ([45]) note that autonomy in this context does not imply full independence from external influence, but rather the perception that one retains ultimate authority in decision making.

Understanding perceived autonomy is essential, as it plays a key role in shaping consumer trust, satisfaction, and usage intention. When consumers feel that they have decision-making agency, they are more likely to embrace AI-based services ([1]; [3]; [7]). Autonomy enhances the perceived role of AI as a supportive aid rather than a controlling agent ([37]). From the perspective of Self-Determination Theory (SDT), autonomy is a fundamental psychological need that fosters intrinsic motivation and engagement ([49]). Importantly, autonomy is not static. Rather than being an inherent individual trait, it is dynamically shaped by the design of the system and the extent to which it enables users to retain control over choices.

Closely related to autonomy is the concept of perceived control, which refers to one’s confidence in their ability to influence outcomes ([8]; [10]). While these concepts are interconnected, they are not interchangeable. Autonomy emphasizes freedom of choice, whereas control pertains to outcome certainty. This distinction becomes clear in contexts like gambling, where an “illusion of control” may lead to overestimations of outcome predictability ([21]; [33]).

Perceived autonomy also contributes to consumer well-being ([17]). Despite its importance, relatively few studies have examined how design elements of AI services can enhance or undermine autonomy perceptions. Some studies point to a trade-off between autonomy and trust—where consumers may accept reduced autonomy if they trust the AI’s competence ([24]). However, for long-term engagement, perceived autonomy remains crucial ([19]).

Autonomy is thus a pivotal determinant of AI service evaluation. This study proposes that perceived autonomy will significantly enhance consumer trust, satisfaction, and usage intention.

Usage intention represents a consumer’s psychological willingness to engage with a technology or service in the future, distinct from initial trials or curiosity-driven usage ([11]; [46]). [6] ([6]) specifically emphasizes continued engagement intentions, highlighting users’ intention to persist with a technology after initial adoption. In the technology acceptance literature, usage intention is viewed as a reliable predictor of actual usage behaviors and is closely associated with consumer trust, satisfaction, and perceived utility ([18]; [26]). Recently, with the proliferation of AI-based services, usage intention has become particularly relevant. [14] ([14]) examined customers’ intentions to use AI-driven investment services (robo-advisors), demonstrating that technology readiness significantly influences intentions. Interestingly, they found technological discomfort positively influenced adoption, suggesting that analytical AI services reduce barriers by placing consumers in relatively passive roles, altering traditional patterns of technology adoption. Similarly, [51] ([51]) explored the continued use intention of smart home assistants, finding that initial gratification significantly predicts sustained engagement. They emphasize usage intention as crucial for assessing long-term success in consumer-focused AI technologies, where user satisfaction drives continuous adoption.

Accordingly, we hypothesize the following:
**H1.** *Perceived autonomy in AI services will positively influence consumer trust.*
**H2.** *Perceived autonomy in AI services will positively influence consumer satisfaction.*
**H3.** *Perceived autonomy in AI services will positively influence consumer usage intention.*

### 2.2. Task Difficulty and Perceived Autonomy in AI Interactions

Task difficulty is a contextual factor that significantly affects consumer-AI interactions. It refers to the perceived difficulty of a task and influences whether consumers seek assistance or resist AI involvement ([9]). High task difficulty may increase reliance on AI, but can also heighten consumer sensitivity to perceived autonomy, especially when AI systems act in dominant roles ([16]).

Consumers often prefer AI to serve as a facilitator rather than a decision maker, particularly when tasks are cognitively demanding. Excessive automation in difficult tasks may reduce perceived autonomy and trigger disengagement ([16]). [9] ([9]) show that when AI becomes too dominant in essential decisions, users may resist its guidance, especially if the stakes are high or the outcomes are personally meaningful. In contrast, consumers are more likely to defer to AI suggestions in low-difficulty, routine decisions where the perceived cost of error is low ([29]).

Further, [40] ([40]) find that transparency in complex AI decisions can backfire by increasing cognitive burden. [25] ([25]) also highlight that task difficulty can amplify the need for explainability and meaningful user involvement, reinforcing the idea that consumers are most comfortable when AI offers guidance but leaves final decisions to them. Moreover, when AI services include collaborative elements—such as shared decision-making responsibility ([36]) or interactive feedback mechanisms ([4])—users are more likely to experience enhanced autonomy and maintain trust, even in high-difficulty settings. [34] ([34]) similarly demonstrate that allowing users to control the detail and timing of explanations helps reduce cognitive burden and improve perceived competence of the AI system.

This interplay between task difficulty and autonomy has implications for engagement. When autonomy is preserved in complex tasks, consumers are more likely to engage positively with AI. Conversely, perceived overreach by AI may reduce trust and discourage use. Therefore, we propose that task complexity moderates the relationship between autonomy and engagement outcomes.

**H4.** 
*Task difficulty will moderate the relationship between perceived autonomy and consumer trust, such that the positive effect of autonomy on trust will be stronger under high task difficulty than under low task complexity.*


**H5.** 
*Task difficulty will moderate the relationship between perceived autonomy and consumer satisfaction, such that the positive effect of autonomy on satisfaction will be stronger under high task difficulty than under low task complexity.*


**H6.** 
*Task difficulty will moderate the relationship between perceived autonomy and consumer usage intention, such that the positive effect of autonomy on usage intention will be stronger under high task difficulty than under low task difficulty.*


### 2.3. AI Design Elements Supporting Perceived Autonomy

AI service design plays a pivotal role in shaping perceived autonomy, particularly through mechanisms such as explanation, feedback, and shared responsibility. While autonomy can be bolstered by system transparency ([23]), explanation alone is not always sufficient. [13] ([13]) argue that relational transparency—rooted in reciprocal interactions rather than one-way disclosures—is more effective in fostering trust and autonomy. Likewise, feedback mechanisms are influential only when they empower users to influence outcomes, rather than merely collecting data ([41]).

Furthermore, shared responsibility frameworks can reinforce autonomy by emphasizing users’ active role in AI-assisted decision making ([36]). When consumers perceive that they are co-agents in the process rather than passive recipients of algorithmic decisions, autonomy is strengthened and satisfaction increases ([4]).

However, these elements are not universally beneficial. If implemented poorly—e.g., overwhelming users with technical explanations or vague feedback—such features may backfire, diminishing trust and autonomy ([40]). Therefore, understanding the combined effect of these design elements is essential.

This study investigates three AI design elements: explainability, feedback, and shared responsibility. Each is hypothesized to positively impact perceived autonomy, which in turn affects key consumer outcomes.

**H7.** 
*AI explainability will positively influence perceived autonomy.*


**H8.** 
*Feedback mechanisms in AI services will positively influence perceived autonomy.*


**H9.** 
*Shared responsibility frameworks will positively influence perceived autonomy.*


### 2.4. Research Model Overview

The proposed research model integrates the relationships described above. Specifically, it positions perceived autonomy as a key variable influenced by AI design elements (H7–H9) and interacting with task difficulty (H4–H6), which subsequently shapes consumer trust (H1, H4), satisfaction (H2, H5), and usage intention (H3, H6), as illustrated in Figure 1. The model distinguishes between two experimental studies: Study 1 tests the role of task difficulty and autonomy in influencing consumer responses, while Study 2 explores how design features (explanation, feedback, shared responsibility) shape perceived autonomy and, in turn, consumer outcomes. Together, these studies aim to illuminate how AI can be designed to promote meaningful consumer autonomy and engagement.

## 3. Empirical Studies

### 3.1. Study 1: Effects of Perceived Autonomy and Task Difficulty

Study 1 investigated how perceived consumer autonomy influences trust, satisfaction, and usage intention in generative AI services (H1–H3) and whether these effects vary based on task difficulty (H4–H6).

Participants for this study were recruited from Embrain’s panel database, a Macromill Group company that maintains a panel of over 1.5 million registered members in South Korea. The study targeted individuals aged 20 to 50 who had used generative AI services at least once in the past three months. An initial sample of 400 participants was recruited for the survey.

To ensure the validity and reliability of the survey instrument, a pilot study was conducted between 14 and 16 January 2025, with 40 participants (10% of the total sample). Based on participant feedback, minor refinements were made to certain survey items before administering the primary survey between 17 and 21 January 2025. Stratified sampling was used to balance gender and age representation across groups. After removing outliers, the final dataset consisted of 332 participants. Participants provided informed consent and were compensated with KRW 2000 (USD 1~2), a level considered appropriate for online panel engagement (e.g., [28]).

Participants were randomly assigned to one of four experimental conditions in a 2 (Task Difficulty: Low vs. High) × 2 (Perceived Autonomy: Absent vs. Present) between-subjects design. The task involved using a generative AI tool (e.g., ChatGPT-4o, CLOVA X 3.0) to summarize national policies. In the low-difficulty condition, participants extracted five keywords from a short paragraph. In the high-difficulty condition, participants synthesized information from government reports to produce a short essay. Perceived autonomy was manipulated by allowing participants to choose between AI responses or request modifications in the present autonomy conditions.

#### 3.1.1. Measurement and Manipulation Check

Dependent variables included trust (three items, α = 0.91; Gefen et al., 2003), satisfaction (single-item adapted from [43]; single-item measures have been validated in prior consumer research, e.g., [5]), and usage intention (three items, α = 0.87; [47]). All variables were measured using seven-point Likert scales. Also, all survey items used in this study are provided in Appendix A.

To validate the manipulations, task difficulty and perceived autonomy were assessed. Task difficulty was measured using two items, with satisfactory internal consistency (α = 0.75), based on cognitive workload measures ([44]). Perceived autonomy was assessed using three items adapted from [42] ([42]) (α = 0.68).

An independent *t*-test confirmed successful manipulation of task difficulty. Participants perceived tasks as significantly more difficult under the high-difficulty condition (M = 5.84, SD = 0.92) than under the low-difficulty condition (M = 5.32, SD = 1.08), t (330) = −4.76, *p* < 0.001. A two-way ANOVA using perceived autonomy as the dependent variable revealed a significant main effect of autonomy manipulation, F (1, 328) = 37.15, *p* < 0.001, and a significant autonomy × task difficulty interaction, F (1, 328) = 2.13, *p* = 0.018, indicating the autonomy manipulation was more effective under high-difficulty conditions. Specifically, the autonomy manipulation effect was larger and significant under high-difficulty conditions (M_present = 5.38, SD = 0.78 vs. M_absent = 4.96, SD = 0.73; t (172) = 3.70, *p* < 0.001), while smaller and less pronounced under low-difficulty conditions (M_present = 5.34, SD = 0.80 vs. M_absent = 5.20, SD = 0.80; t (156) = 1.08, *p* = 0.28).

#### 3.1.2. Descriptive and Preliminary Analyses

OLS regression analyses were selected as the primary method due to their robustness in testing direct and interaction effects of manipulated experimental variables on dependent outcomes. Correlation analyses were conducted as preliminary assessments to verify relationships among variables, and independent *t*-tests were used to clarify group-level differences based on significant interaction terms.

To confirm random assignment success, chi-square and ANOVA tests showed no significant group differences in gender, age, education, income, AI usage frequency, or digital competence. Descriptive statistics of participants are reported in Table 1.

-Group 1: low task difficulty, absent perceived autonomy-Group 2: low task difficulty, present perceived autonomy-Group 3: high task difficulty, absent perceived autonomy-Group 4: high task difficulty, present perceived autonomy

Bivariate correlations showed that perceived autonomy was significantly correlated with trust (r = 0.52), satisfaction (r = 0.62), and usage intention (r = 0.55), all *p* < 0.001. Task difficulty was weakly but significantly correlated with usage intention (r = 0.31, *p* < 0.001). Table 2 provides the full correlation matrix.

#### 3.1.3. Hypothesis Testing

Ordinary least squares (OLS) regression analyses were conducted to test the main and interaction effects of perceived autonomy and task difficulty. Perceived autonomy significantly predicted trust (B = 0.492, SE = 0.053, *p* < 0.001), satisfaction (B = 0.619, SE = 0.051, *p* < 0.001), and usage intention (B = 0.460, SE = 0.048, *p* < 0.001), supporting H1 through H3. However, the interaction terms between autonomy and task difficulty were not significant across all models, failing to support H4 through H6. Specifically, interaction terms between autonomy and task difficulty were not significant for trust (B = 0.094, *p* = 0.324), satisfaction (B = 0.129, *p* = 0.168), or usage intention (B = 0.143, *p* = 0.101), failing to support H4 through H6. Although the interaction terms (H4–H6) were not statistically significant, subsequent group-level analyses indicated meaningful differences, warranting further exploration in the Discussion section. The full models, including controls and interaction terms, appear in Table 3.

#### 3.1.4. Group-Level Comparisons

To further examine how perceived autonomy functioned within each task difficulty level, independent *t*-tests were conducted. Under the low-difficulty condition (Groups 1 vs. 2), there were no significant differences in trust (t = 1.090, *p* = 0.278), satisfaction (t = 0.951, *p* = 0.343), or usage intention (t = 1.376, *p* = 0.171). However, under the high-difficulty condition (Groups 3 vs. 4), present autonomy participants reported significantly greater trust (M = 5.16 vs. 4.90; t = 2.044, *p* = 0.042) and satisfaction (M = 5.52 vs. 5.20; t = 2.358, *p* = 0.020). Differences in usage intention approached significance (t = 1.928, *p* = 0.056). These findings are visualized in Figure 2.

Overall, these results indicate that perceived autonomy plays a more pronounced role in influencing trust and satisfaction when task difficulty is high. At the same time, its effects are less evident in contexts with low difficulty. This suggests that AI services designed for complex decision making should prioritize autonomy-enhancing features to build user trust and satisfaction, whereas, for more straightforward tasks, usability and efficiency may be more critical than autonomy.

#### 3.1.5. Discussion

Study 1 provides strong support for the importance of perceived autonomy in shaping consumer responses to AI services. Autonomy positively influenced trust, satisfaction, and usage intention, confirming hypotheses H1 through H3. These effects were particularly salient in high-complexity contexts, where participants reported greater benefits from autonomy-supportive features.

The pronounced effect of perceived autonomy in high-complexity contexts suggests that consumers rely more heavily on autonomy-supportive AI features when task demands exceed their cognitive resources, aligning with prior theories emphasizing autonomy as critical for reducing anxiety and enhancing efficacy in complex tasks ([19]).

Although interaction effects (H4 through H6) were not statistically significant in the regression models, the group-level comparisons suggest a meaningful pattern: autonomy had stronger effects when task complexity was high. This partial alignment with the hypothesized moderation indicates that while the statistical interaction was not significant, the direction of effects supports the theoretical premise.

Unlike [38]’s ([38]) generalized autonomy hypothesis, our findings specifically highlight task-dependent variability in autonomy’s effectiveness, reinforcing [19]’s ([19]) assertion that autonomy’s value in technology interactions intensifies under conditions of uncertainty or complexity.

Service providers should prioritize autonomy-enhancing features, such as offering customizable response options or easy modification pathways, especially for high-complexity scenarios like financial planning or medical decision making, to significantly enhance consumer trust and satisfaction. Interestingly, the stronger autonomy manipulation effect in high-difficulty tasks aligns with prior research (e.g., [19]), suggesting that autonomy becomes particularly salient when cognitive demands are high, as users actively seek greater control and decision-making support.

Study 1 has several limitations. First, the use of a single-item measure for satisfaction may limit construct validity. Future research should employ multi-item scales for robustness. Second, the cross-sectional design does not fully address causal directionality; thus, longitudinal studies would be beneficial. Lastly, the significant interaction observed in the autonomy manipulation suggests the effectiveness of the manipulation varied with task difficulty. Although effective overall, the weaker differentiation under low-difficulty conditions indicates a methodological limitation. Future studies could refine autonomy manipulations or standardize tasks to ensure consistent effects.

### 3.2. Study 2

Study 1 demonstrated that perceived consumer autonomy plays a critical role in shaping consumer trust, satisfaction, and continued usage intention in AI services, with its influence becoming more pronounced under high task difficulty conditions. These findings suggest that autonomy is not merely a passive perception but an essential determinant of consumer engagement with AI services, particularly in complex decision-making contexts. However, Study 1 examined autonomy as a naturally occurring perception within structured scenarios rather than an actively modifiable variable.

Building on these insights, Study 2 investigates whether perceived consumer autonomy can be intentionally influenced through specific AI design interventions. Drawing from prior research that has identified factors affecting autonomy, this study examines the effectiveness of three targeted design elements—explainability, feedback, and shared responsibility—in modifying consumer perceptions of autonomy. These interventions are designed to test whether transparency (explainability), user engagement (feedback), and explicit reinforcement of decision-making responsibility (shared responsibility) can shape consumer autonomy and, in turn, impact trust, satisfaction, and continued usage intention.

By implementing these interventions, Study 2 seeks to determine under what conditions autonomy perceptions can be strengthened and whether these changes lead to improved consumer responses. If effective, these findings will provide practical insights for developing AI services that provide information and actively empower users in decision-making processes. Furthermore, this study expands the understanding of how AI services can be optimized to foster greater consumer trust and long-term engagement by systematically evaluating the mechanisms that drive perceived autonomy in AI interactions.

#### 3.2.1. Participants

Study 2 recruited participants using the same sampling strategy as Study 1, drawing from Embrain’s panel database, which includes over 1.5 million registered panel members in South Korea. The study targeted individuals aged 20 to 50 who had used generative AI services at least once in the past three months. To ensure a balanced representation across gender and age groups, stratified sampling methods were applied, mirroring Study 1. Additionally, participants in Study 1 were excluded from Study 2 to avoid potential carryover effects.

Before the primary survey, a pilot study was conducted from January 17 to 21 with 40 participants (approximately 10% of the target sample) to refine survey items and scenario instructions. The primary survey was conducted between 22 and 24 January, and after removing outliers, the final sample consisted of 376 participants.

As in Study 1, participants were given detailed information about the study and were required to click the “consent and proceed” button before participating. They were also informed that they could withdraw without penalty and compensated with KRW 2000 (USD 1~2), a level considered appropriate for online panel engagement (e.g., [28]).

#### 3.2.2. Design and Procedure

Study 2 investigated whether perceived consumer autonomy could be actively modified through specific AI design interventions, including explainability, feedback, and shared responsibility. A between-subjects experimental design was employed, with participants randomly assigned to one of four conditions.

-Control condition (Group 1): The AI service provided a standard response with no additional design interventions.-Explainability condition (Group 2): The AI explained the reasoning behind its response.-Feedback condition (Group 3): Participants were asked to justify their response selection, with the AI informing them that their feedback would be used to improve future AI outputs.-Shared responsibility condition (Group 4): Participants were explicitly informed that the AI service’s response was for reference only and that they held the final responsibility for evaluating and applying the information.

Each participant was given a task that required them to summarize recent government policies addressing aging populations using a generative AI service. The AI system presented two different response styles, and participants had to choose their preferred response. Depending on the condition, they received additional explanations, provided feedback, or were reminded of their decision-making responsibility.

Upon completing the primary task, participants answered a series of questions about their assigned scenario. An example of the scenario provided to participants is available in Appendix B. The demographic characteristics of participants across the four conditions were analyzed, and the results are presented in Table 4. No significant differences were found in gender, age, education level, or prior AI usage frequency, confirming that randomization was successful.

#### 3.2.3. Measures

Measures for perceived autonomy, trust, satisfaction, and usage intention were identical to those used in Study 1. Perceived autonomy was measured using three items ([42], α = 0.713). Trust was measured using three items ([18], α = 0.908). Satisfaction was measured by a single-item scale adapted from [43] ([43]). Usage intention was measured with three items adapted from [47] ([47], α = 0.889). All items were assessed using seven-point Likert scales (1 = strongly disagree, 7 = strongly agree), and full survey items are provided in Appendix A.

#### 3.2.4. Manipulation Check

Manipulation checks were conducted to verify that the design interventions were successfully implemented. It used one-way ANOVAs to confirm the successful implementation of each AI design element across conditions. The results (Table 5) confirmed that participants in the treatment conditions perceived significant differences in the levels of explainability, feedback, and shared responsibility provided by the AI service.

Post-hoc analyses (Duncan’s test) indicated that perceived explainability was significantly higher in Group 2 (explainability condition; M = 5.24, SD = 0.81) compared to Groups 1, 3, and 4 (*p* < 0.05). Similarly, perceived feedback was significantly greater in Group 3 (feedback condition; M = 4.78, SD = 1.25) than in Groups 1, 2, and 4 (*p* < 0.05). Lastly, perceived shared responsibility was significantly higher in Group 4 (shared responsibility condition; M = 5.29, SD = 1.21) compared to Groups 1, 2, and 3 (*p* < 0.05).

These results confirm that the manipulations for explainability, feedback, and shared responsibility were successfully implemented, ensuring that participants in each treatment condition experienced distinct levels of each factor.

#### 3.2.5. Results

In addition to the manipulation check, Figure 3 compares perceived consumer autonomy, trust, satisfaction, and continued usage intention across the four experimental conditions. While significant differences in manipulation perception were observed, no statistically significant group differences were found in perceived consumer autonomy (F = 1.174, *p* > 0.05), trust (F = 1.339, *p* > 0.05), satisfaction (F = 2.465, *p* > 0.05), or continued usage intention (F = 0.320, *p* > 0.05). However, subsequent regression analyses revealed meaningful relationships at the individual level, suggesting potential within-group variations not captured by simple group comparisons.

This suggests that while participants could recognize differences in AI design elements, these differences did not translate into substantial variations in autonomy perception or behavioral outcomes at the group level. However, as prior research suggests ([23]), autonomy may not always be directly influenced by transparency and feedback mechanisms but rather by more profound engagement with AI decision-making processes. Thus, to further explore the effects of these design elements on autonomy and user engagement, individual-level variations will be analyzed using multiple regression models presented in the following section.

A Pearson correlation analysis examined the relationships among perceived consumer autonomy, trust, satisfaction, and continued usage intention. The results, summarized in Table 6, confirm that perceived autonomy was positively and significantly correlated with trust (r = 0.631, *p* < 0.001), satisfaction (r = 0.600, *p* < 0.001), and continued usage intention (r = 0.551, *p* < 0.001).

These findings indicate that consumers who perceive higher levels of autonomy in AI interactions are more likely to develop stronger trust, experience higher satisfaction, and demonstrate greater willingness to continue using similar AI services. Furthermore, trust exhibited a strong positive correlation with both satisfaction (r = 0.670, *p* < 0.001) and continued usage intention (r = 0.588, *p* < 0.001), reinforcing the role of trust as a key determinant of consumer engagement in AI-mediated decision making. Satisfaction also showed a robust correlation with continued usage intention (r = 0.650, *p* < 0.001), suggesting that higher consumer satisfaction is closely linked to the long-term adoption of AI services.

Multiple regression analyses were conducted for each experimental condition to further analyze the impact of perceived consumer autonomy. Table 7 presents the regression results.

The regression analysis results reveal significant insights into the role of perceived consumer autonomy and trust in shaping consumer responses to AI services. Perceived autonomy emerged as a strong predictor of trust across all experimental groups. This effect was most pronounced in the shared responsibility condition (*b* = 0.372, *p* < 0.001), suggesting that AI services that actively engage users in decision making enhance perceptions of autonomy and trust. In contrast, the explainability and feedback conditions exhibited weaker effects, indicating that merely providing additional information does not necessarily strengthen perceived autonomy.

The influence of perceived autonomy on satisfaction and continued usage intention varied depending on the experimental condition. Autonomy significantly predicted satisfaction in Groups 1, 3, and 4. In contrast, it did not show a significant effect in the explainability condition (*b* = 0.058, *p* > 0.05). Similarly, for continued usage intention, autonomy played a substantial role in Groups 3 and 4 (*b* = 0.349, *p* < 0.01 in Group 3; *b* = 0.315, *p* < 0.01 in Group 4), whereas its effects were not statistically significant in Groups 1 and 2. These findings indicate that while autonomy positively influences consumer engagement, its impact is contingent upon the nature of the AI design intervention.

Trust was consistently a strong predictor of satisfaction and continued usage intention across all conditions. It significantly influenced satisfaction in all groups (*b* = 0.452, *p* < 0.001 in Group 1; *b* = 0.685, *p* < 0.001 in Group 2; *b* = 0.411, *p* < 0.001 in Group 3; *b* = 0.426, *p* < 0.001 in Group 4). Additionally, trust played a key role in driving continued usage intention in Groups 1, 3, and 4 (*b* = 0.457, *p* < 0.001 in Group 1; *b* = 0.319, *p* < 0.001 in Group 3; *b* = 0.401, *p* < 0.001 in Group 4), but its effect was relatively weaker under the explainability condition (*b* = 0.343, *p* < 0.05). These findings reinforce the critical role of trust in fostering consumer satisfaction and long-term engagement with AI services.

Overall, these results suggest that shared responsibility is the most effective AI design element for enhancing consumer autonomy, strengthening trust, satisfaction, and continued engagement. In contrast, explainability and feedback alone did not significantly increase perceived autonomy, satisfaction, or long-term engagement. These findings emphasize the importance of AI design strategies involving users in decision-making rather than solely focusing on information transparency or passive feedback mechanisms.

Taken together, the results clarify the status of hypotheses H7 through H9 proposed in this study. Specifically, the explainability condition (H7) did not significantly enhance perceived autonomy compared to the control condition; therefore, H7 was not supported. Similarly, the feedback condition (H8) also failed to yield a significant positive effect on perceived autonomy, leading to the conclusion that H8 was not supported. In contrast, the shared responsibility condition (H9) demonstrated a significant positive impact on perceived autonomy, confirming that H9 was supported.

#### 3.2.6. Discussion

Study 2 examined whether perceived autonomy can be actively shaped through AI design interventions—specifically, explainability, feedback, and shared responsibility. The results reinforce the role of autonomy in fostering trust and engagement but suggest that not all design elements are equally effective in enhancing autonomy perceptions.

Among the three interventions, shared responsibility had the most significant effect on autonomy and trust. This supports prior research indicating that actively involving users in AI-driven decisions enhances perceptions of control and trust ([15]). In contrast, explainability and feedback alone did not significantly strengthen autonomy perceptions. The lack of a strong effect from explainability aligns with findings suggesting that merely providing more information does not necessarily improve perceived autonomy and, in some cases, may lead to information overload ([23]). Similarly, the limited impact of feedback suggests that users may not feel a greater sense of autonomy unless they perceive their input as directly shaping AI-generated outcomes ([20]). These findings align with prior research indicating that while explainability and feedback mechanisms may enhance trust, they do not necessarily strengthen perceived autonomy unless users feel that their actions have a meaningful impact on AI-generated recommendations. Effective autonomy-enhancing design requires interactive mechanisms that allow users to influence AI decision making rather than relying solely on passive transparency ([41]).

Trust independently emerged as a significant predictor of satisfaction and usage intention, highlighting its standalone importance in consumer responses. In particular, trust strongly influenced satisfaction in the explainability condition, even when autonomy did not significantly increase. This suggests that transparency independently enhances trust. However, it does not automatically enhance autonomy, which appears contingent on meaningful user engagement. These findings underscore the importance of designing AI systems that provide information and ensure users feel actively involved in decision making.

From a practical standpoint, these findings suggest that shared responsibility mechanisms—such as allowing users to refine AI-generated outputs or explicitly acknowledge their role in final decisions—reinforce autonomy, trust, and engagement. Meanwhile, explainability and feedback should be designed carefully to prevent cognitive overload and ensure that users perceive their input as meaningful.

Together, these findings suggest that autonomy-enhancing design is not a one-size-fits-all solution; AI services must be tailored to balance transparency, user control, and automation effectively. The General Discussion further explores the broader implications of these findings for AI trust, engagement, and consumer well-being.

Although manipulation checks were statistically significant, some effect sizes, particularly for explainability, were relatively small. This could partly explain the lack of strong autonomy effects observed under the explainability condition. It should be noted that the current design independently manipulated explainability, feedback, and shared responsibility. Future studies might benefit from exploring the combined or interactive effects of these AI design elements, which may provide deeper insights into their synergistic roles.

## 4. General Discussion

This research investigated the role of perceived autonomy in AI-assisted decision-making and its impact on trust, satisfaction, and usage intention. Study 1 explored how task difficulty moderates the relationship between autonomy and consumer responses. Study 2 examined whether specific AI design elements—explainability, feedback, and shared responsibility—can actively enhance perceived autonomy and subsequently influence user engagement. Collectively, these studies provide an integrated understanding of autonomy’s role in AI-mediated interactions and offer practical insights into optimizing consumer experiences with AI services.

One key finding is that perceived autonomy significantly enhances trust in AI interactions, confirming prior evidence that autonomy serves as a critical factor in fostering consumer confidence in technology-driven decisions ([15]; [53]). Study 1 demonstrated this relationship is particularly pronounced in high-complexity tasks, suggesting that consumers value autonomy more when facing cognitively demanding decisions ([29]). However, the influence of autonomy on satisfaction and usage intention appeared context dependent. In Study 2, only shared responsibility significantly improved satisfaction and engagement. These results suggest that to effectively enhance consumer engagement, perceived autonomy must be coupled with active user involvement rather than passive interactions with AI-generated outputs.

Another significant contribution is the identification that not all AI design elements equally enhance autonomy perceptions. Specifically, Study 2 revealed that among the tested interventions, only shared responsibility consistently and meaningfully increased autonomy perceptions and subsequent engagement. This result refines previous assumptions that transparency alone sufficiently improves user control perceptions, highlighting instead the importance of actively involving users in decision-making processes ([7]). In contrast, explainability and feedback, although conceptually related to autonomy, showed limited effectiveness when implemented alone. These findings align with prior literature, suggesting excessive information may lead to cognitive overload, diminishing perceived autonomy ([23]). Similarly, passive feedback mechanisms were not perceived as autonomy enhancing unless users clearly saw their input directly impacting AI-generated results ([20]).

Trust independently emerged as a strong predictor of satisfaction and usage intention across various AI design conditions. Particularly under the explainability condition of Study 2, trust significantly influenced satisfaction even without corresponding increases in autonomy. This underscores the distinct and critical role of trust in user experiences with AI, independent of autonomy perceptions. Prior research warns against automation bias, where excessive reliance on AI recommendations can reduce perceived autonomy ([20]). These findings emphasize designing AI systems that enhance trust while simultaneously preserving user control to support, rather than replace, consumer judgment.

From a practical perspective, this research offers crucial recommendations for AI developers and policymakers. AI services should be designed as interactive decision-support tools rather than passive recommendation engines. Shared responsibility emerged as the most effective approach, suggesting AI services should explicitly position users as active decision makers through mechanisms such as iterative refinement options, multi-step decision processes, and explicit user acknowledgment of final decision authority.

Explainability alone should not be considered sufficient for enhancing perceived autonomy. While transparency remains essential for trust building, it needs to be combined with interactive control mechanisms enabling users to actively engage with and modify AI-generated recommendations ([19]). Feedback mechanisms should also clearly reflect user influence, ensuring user input visibly shapes AI outcomes rather than serving merely as passive data collection. Without meaningful user engagement, feedback may appear superficial and insufficient to enhance autonomy.

Several important future research directions emerge from this study. Firstly, future studies should examine how individual differences in AI literacy, cognitive styles, and decision-making preferences moderate responses to autonomy-enhancing AI designs. Understanding these nuances can help personalize AI services effectively ([52]). Secondly, while this research focused on immediate perceptions of autonomy, longitudinal studies are necessary to assess whether initial autonomy enhancements lead to sustained trust and engagement over time. Lastly, exploring regulatory and ethical frameworks to ensure AI services effectively balance automation and user agency is critical for responsible AI development ([35]).

Addressing these areas can significantly contribute to developing AI services that not only enhance efficiency but also actively reinforce consumer autonomy, trust, and sustained engagement. As AI continues to permeate daily life, ensuring consumer control and autonomy remains essential for ethical and human-centered AI deployment.

## 5. Conclusions

This research examined the role of perceived consumer autonomy in shaping trust, satisfaction, and usage intention in AI services. Across two experimental studies, this research provides empirical evidence that autonomy significantly enhances consumer trust and engagement in AI interactions. Study 1 showed that autonomy exerts stronger positive effects in high-complexity tasks, where consumers actively seek greater control over AI-assisted decisions. Study 2 demonstrated that not all AI design interventions effectively reinforce perceived autonomy. Among the tested interventions, only shared responsibility significantly increased perceptions of autonomy, trust, and long-term engagement, whereas explainability and feedback alone were insufficient.

These findings contribute meaningfully to understanding consumer autonomy in AI interactions. First, the research empirically confirms that perceived autonomy is a fundamental determinant of consumer trust and satisfaction with AI services. Consumers who feel greater autonomy when interacting with AI exhibit higher trust, greater satisfaction, and stronger intentions to continue using these services. Second, this study identifies shared responsibility as the most effective AI design element for enhancing consumer autonomy, distinguishing it from less effective interventions such as explainability or feedback alone. These insights extend prior knowledge by highlighting that merely providing transparency or soliciting user feedback does not automatically strengthen perceptions of autonomy unless accompanied by active consumer engagement.

From a practical standpoint, the findings suggest that AI services should be designed as interactive decision-support tools rather than passive recommendation systems. To empower consumers effectively, AI systems must incorporate mechanisms that encourage active participation, such as iterative refinements of AI-generated content, explicit acknowledgment of user decision-making responsibility, and opportunities to customize outputs. Transparency alone is insufficient; explainability features should be combined with meaningful user control options. Similarly, feedback mechanisms should enable users to see tangible effects of their input, thereby reinforcing genuine perceptions of autonomy and active engagement.

Despite these valuable contributions, several critical areas remain open for future research. Subsequent studies should explore how individual differences such as AI literacy, cognitive load, and decision-making styles influence consumer responses to autonomy-enhancing AI designs. Additionally, research should investigate the long-term impact of autonomy-supportive AI services on consumer trust, satisfaction, and usage intention, particularly in sustained, real-world decision-making contexts. Finally, regulatory and ethical considerations must be examined to ensure AI systems maintain an appropriate balance between automation and user agency, promoting responsible AI development.

As AI technologies increasingly integrate into consumers’ daily lives, preserving consumer autonomy will be essential to foster trust, engagement, and ethical deployment. This research emphasizes that autonomy-enhancing design in AI is not merely an additional benefit but an indispensable foundation for consumer-oriented, responsible AI services.

## Figures and Tables

**Figure 1 behavsci-15-00534-f001:**
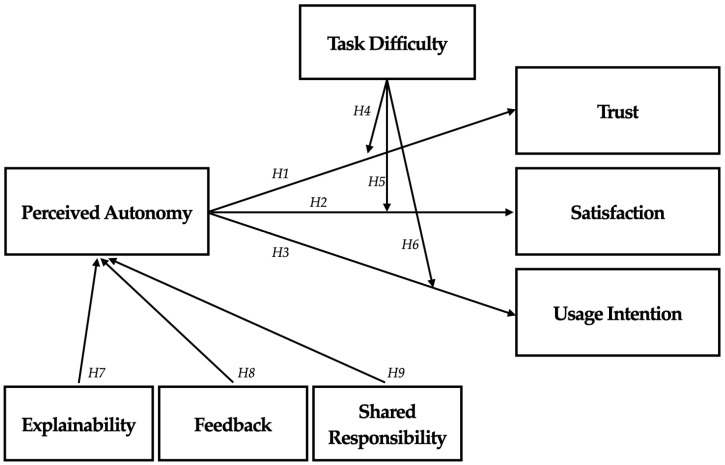
Research model.

**Figure 2 behavsci-15-00534-f002:**
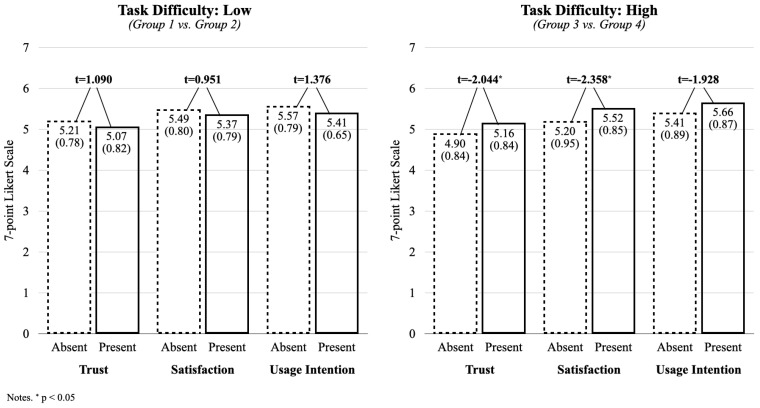
Group-level comparisons of dependent variables.

**Figure 3 behavsci-15-00534-f003:**
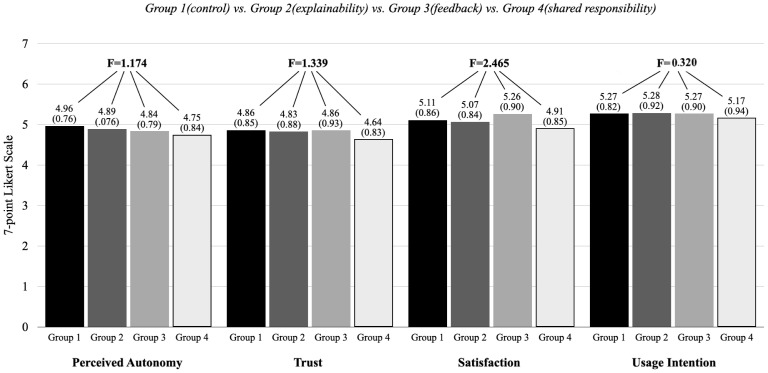
Comparison of perceived autonomy, trust, satisfaction, and usage intention across experimental conditions.

**Table 1 behavsci-15-00534-t001:** Descriptive statistics of participants.

	Total Sample (N = 332)	Group 1 (N = 90)	Group 2 (N = 68)	Group 3 (N = 87)	Group 4 (N = 87)	X^2^/F
Gender (%)						0.749
Male	38.3	38.9	33.8	39.1	40.2	
Female	61.7	61.1	66.2	60.9	59.8	
Age (M, SD)	38.57 (11.03)	38.82 (10.81)	37.69 (11.87)	38.61 (10.60)	38.95 (11.17)	0.194
Education (%)						7.585
High school graduate	19.3	12.2	26.5	23.0	17.2	
College graduate	70.8	74.4	63.2	69.0	74.7	
Postgraduate degree	9.9	13.3	10.3	8.0	8.0	
Household income (KRW 10k, M, SD)	327.94 (202.11)	339.40 (177.67)	286.28 (178.74)	334.34 (210.82)	342.24 (231.20)	1.237
Freq. of generative AI usage (%)						14.331
Several times a day	25.9	22.2	26.5	27.6	27.6	
Once a day (daily)	10.2	11.1	8.8	8.0	12.6	
5–6 times a week	7.5	5.6	5.9	6.9	11.5	
3–4 times a week	15.1	22.2	11.8	14.9	10.3	
1–2 times a week	20.5	18.9	17.6	23.0	21.8	
1–3 times a month	15.4	15.6	22.1	12.6	12.6	
Less than once a month	5.4	4.4	7.4	6.9	3.4	
Objective competence (Generative AI, M, SD)	4.27 (1.26)	4.24 (1.32)	4.15 (1.18)	4.32 (1.31)	4.36 (1.21)	0.413
Subjective competence (Generative AI, M, SD)	5.39 (0.78)	5.42 (0.76)	5.25 (0.80)	5.39 (0.78)	5.46 (0.80)	0.931

Notes: M = mean; SD = standard deviation; N = number of observations.

**Table 2 behavsci-15-00534-t002:** Correlations among task difficulty, perceived autonomy, and dependent variables.

	Task Difficulty	Perceived Autonomy	Trust	Satisfaction	Usage Intention
Task difficulty	1.000				
Perceived autonomy	0.242 ***	1.000			
Trust	0.156 **	0.520 ***	1.000		
Satisfaction	0.165 **	0.619 ***	0.692 ***	1.000	
Usage intention	0.313 ***	0.554 ***	0.550 ***	0.687 ***	1.000

*** *p* < 0.001, ** *p* < 0.01.

**Table 3 behavsci-15-00534-t003:** Regression results for trust, satisfaction, and usage intention.

	-						With Interaction Terms					
	b	SE	b	SE	b	SE	b	SE	b	SE	b	SE
Dependent Variable	Trust		Satisfaction		Usage Intention		Trust		Satisfaction		Usage Intention	
Intercept	1.056 **	0.398	1.204 **	0.388	1.085 **	0.364	1.348 **	0.478	1.577 **	0.467	1.473 **	0.437
Gender (male = 0)	−0.115	0.083	−0.030	0.081	−0.073	0.076	−0.115	0.083	−0.029	0.081	−0.070	0.076
Age	0.013 **	0.004	0.002	0.004	0.006	0.004	0.014 **	0.004	0.002	0.004	0.006	0.004
Monthly income (KRW 10k)	−0.000	0.000	0.000	0.000	0.000	0.000	−0.000	0.000	0.000	0.000	0.000	0.000
High school (university = 0)	−0.014	0.105	−0.020	0.102	0.060	0.096	−0.006	0.105	−0.011	0.102	0.066	0.096
Graduate (university = 0)	0.053	0.128	0.073	0.125	−0.136	0.117	0.039	0.128	0.061	0.125	−0.141	0.117
Daily usage (week = 0)	−0.061	0.086	0.034	0.084	0.308 ***	0.079	−0.059	0.086	0.035	0.084	0.308 ***	0.079
Monthly usage (week = 0)	0.038	0.102	−0.107	0.099	−0.040	0.093	0.040	0.102	−0.099	0.100	−0.028	0.093
Objective competence	−0.013	0.032	−0.038	0.031	0.053	0.029	−0.013	0.032	−0.040	0.031	0.049	0.029
Subjective competence	0.211 ***	0.053	0.193 ***	0.052	0.272 ***	0.049	0.215 ***	0.054	0.195 ***	0.052	0.272 ***	0.049
Perceived autonomy	0.492 ***	0.053	0.619 ***	0.051	0.460 ***	0.048	0.439 ***	0.072	0.551 ***	0.070	0.388 ***	0.066
Task difficulty (low = 0)							−0.573	0.503	−0.716	0.490	−0.729	0.459
Difficulty × autonomy							0.094	0.095	0.129	0.093	0.143	0.087
F	16.397 ***		23.150 ***		25.101 ***		13.862 ***		19.499 ***		21.193 ***	
Adj. R^2^	0.318		0.401		0.421		0.318		0.401		0.423	

Notes. Coefficients represent unstandardized regression estimates (b). Standard errors are in parentheses. *** *p* < 0.001, ** *p* < 0.01.

**Table 4 behavsci-15-00534-t004:** Descriptive statistics of participants.

	Total Sample (N = 376)	Group 1 (N = 95)	Group 2 (N = 96)	Group 3 (N = 94)	Group 4 (N = 91)	X^2^/F
Gender (%)						2.971
Male	40.2	45.3	35.4	43.6	36.3	
Female	59.8	54.7	64.6	56.4	63.7	
Age (M, SD)	39.40 (10.77)	39.61 (11.06)	39.16 (10.74)	39.52 (10.06)	39.32 (11.37)	0.034
Education (%)						12.743 *
High school graduate	12.5	10.5	17.7	9.6	12.1	
College graduate	73.9	72.6	70.8	70.2	82.4	
Postgraduate degree	13.6	16.8	11.5	20.2	5.5	
Household income (KRW 10k, M, SD)	337.16 (198.55)	347.05 (218.17)	316.13 (170.06)	347.80 (186.25)	338.05 (217.89)	0.526
Freq. of generative AI usage (%)						22.539
Several times a day	26.6	27.4	19.8	28.7	30.8	
Once a day (daily)	7.7	10.5	8.3	6.4	5.5	
5–6 times a week	6.9	9.5	5.2	6.4	6.6	
3–4 times a week	20.7	21.1	25.0	21.3	15.4	
1–2 times a week	19.1	20.0	18.8	11.7	26.4	
1–3 times a month	14.4	10.5	17.7	20.2	8.8	
Less than once a month	4.5	1.1	5.2	5.3	6.6	
Objective competence (generative AI, M, SD)	4.28 (1.25)	4.52 (1.15)	4.17 (1.32)	4.12 (1.30)	4.31 (1.19)	1.960
Subjective competence (generative AI, M, SD)	5.25 (0.74)	5.31 (0.68)	5.17 (0.70)	5.28 (0.80)	5.25 (0.79)	0.574

Notes. M = mean; SD = standard deviation; N = number of observations. * *p* < 0.05.

**Table 5 behavsci-15-00534-t005:** Manipulation check results.

	Explainability	Feedback	Shared Responsibility
Group 1 (n = 95)	5.11 _a_ (0.91)	4.38 _a_ (1.41)	4.05 _b_ (1.53)
Group 2 (n = 96)	5.24 _b_ (0.81)	4.42 _a_ (1.31)	4.13 _b_ (1.35)
Group 3 (n = 94)	4.93 _a_ (1.01)	4.78 _b_ (1.25)	3.63 _a_ (1.58)
Group 4 (n = 91)	4.86 _a_ (0.99)	4.02 _a_ (1.48)	5.29 _c_ (1.21)

Notes: M = mean; SD = standard deviation (in parentheses). Means with different subscripts (a–c) differ significantly at *p* < 0.05 according to Duncan’s multiple range test.

**Table 6 behavsci-15-00534-t006:** Correlations between perceived consumer autonomy and dependent variables.

	Perceived Consumer Autonomy	Trust	Satisfaction	Continued Usage Intention
Perceived consumer autonomy	1.000			
Trust	0.631 ***	1.000		
Satisfaction	0.600 ***	0.670 ***	1.000	
Continued usage intention	0.551 ***	0.588 ***	0.650 ***	1.000

*** *p* < 0.001.

**Table 7 behavsci-15-00534-t007:** Regression results for trust, satisfaction, and usage intention.

Dependent Variable	Group 1 (Control)			Group 2 (Explainability)			Group 3 (Feedback)			Group 4 (Shared Responsibility)		
	Trust	Satisfaction	Usage Intention	Trust	Satisfaction	Usage Intention	Trust	Satisfaction	Usage Intention	Trust	Satisfaction	Usage Intention
Intercept	0.708 (0.816)	0.745 (0.850)	1.453 (0.839)	0.174 (0.684)	1.995 ** (0.732)	1.787 (0.945)	−0.228 (0.662)	1.147 * (0.530)	1.165 * (0.537)	1.363 (0.793)	0.809 (0.692)	−0.003 (0.771)
Gender (male = 0)	−0.231 (0.145)	−0.151 (0.153)	0.009 (0.151)	−0.423 ** (0.147)	0.044 (0.165)	−0.050 (0.213)	0.086 (0.160)	−0.185 (0.128)	−0.132 (0.130)	−0.198 (0.162)	0.168 (0.140)	0.282 (0.156)
Age	0.013 (0.008)	0.005 (0.008)	−0.008 (0.008)	0.002 (0.007)	−0.012 (0.007)	−0.013 (0.009)	0.017 (0.009)	−0.016 * (0.008)	−0.009 (0.008)	0.020 * (0.008)	0.000 (0.007)	0.007 (0.008)
Monthly income (KRW 10k)	−0.001 (0.000)	−0.000 (0.000)	0.000 (0.000)	−0.000 (0.000)	0.000 (0.000)	0.001 (0.001)	0.001 (0.001)	0.000 (0.000)	−0.000 (0.000)	−0.001 (0.000)	−0.001 (0.000)	0.000 (0.000)
High school (university = 0)	−0.240 (0.257)	−0.396 (0.268)	−0.016 (0.265)	0.092 (0.167)	0.045 (0.179)	−0.082 (0.231)	−0.143 (0.258)	0.096 (0.207)	−0.065 (0.210)	−0.121 (0.245)	−0.468 * (0.211)	−0.104 (0.235)
Graduate (university = 0)	−0.024 (0.188)	−0.262 (0.195)	−0.090 (0.192)	0.269 (0.181)	−0.005 (0.197)	−0.163 (0.254)	−0.149 (0.189)	0.283 (0.152)	0.315 * (0.154)	0.338 (0.345)	0.294 (0.297)	0.313 (0.331)
Week usage (daily = 0)	−0.259 (0.156)	0.108 (0.164)	−0.035 (0.162)	0.068 (0.147)	−0.069 (0.157)	−0.184 (0.203)	−0.040 (0.176)	−0.067 (0.141)	−0.184 (0.143)	0.009 (0.170)	0.008 (0.146)	−0.176 (0.163)
Monthly usage (daily = 0)	−0.068 (0.237)	0.191 (0.246)	−0.212 (0.243)	0.079 (0.172)	−0.251 (0.184)	−0.179 (0.238)	−0.064 (0.194)	−0.051 (0.155)	−0.393 * (0.157)	0.008 (0.250)	0.140 (0.214)	−0.340 (0.238)
Objective competence	0.032 (0.063)	0.115 (0.066)	0.037 (0.065)	−0.045 (0.045)	0.009 (0.048)	0.130 * (0.062)	−0.021 (0.060)	0.093 (0.048)	0.161 ** (0.049)	−0.077 (0.072)	0.022 (0.062)	−0.004 (0.069)
Subjective competence	0.087 (0.113)	−0.002 (0.118)	0.282 * (0.117)	0.320 ** (0.103)	−0.027 (0.116)	0.089 (0.150)	0.259 * (0.110)	0.042 (0.091)	0.140 (0.092)	0.262 * (0.113)	0.029 (0.100)	0.311 ** (0.112)
Perceived consumer autonomy	0.713 *** (0.105)	0.317 * (0.136)	0.044 (0.134)	0.675 *** (0.093)	0.058 (0.127)	0.273 (0.163)	0.614 *** (0.107)	0.444 *** (0.101)	0.349 ** (0.103)	0.372 *** (0.098)	0.416 *** (0.091)	0.315 ** (0.102)
Trust	-	0.452 *** (0.113)	0.457 *** (0.112)	-	0.685 *** (0.116)	0.343 * (0.150)	-	0.411 *** (0.088)	0.319 *** (0.089)	-	0.426 *** (0.096)	0.401 *** (0.107)
F	8.157 ***	6.514 ***	5.516 ***	16.369 ***	10.322 ***	5.494 ***	8.618 ***	15.289 ***	14.417 ***	4.616 ***	9.316 ***	9.020 ***
Adj. R^2^	0.432	0.392	0.346	0.618	0.519	0.342	0.450	0.628	0.613	0.287	0.504	0.495

Notes. Coefficients represent unstandardized regression estimates (b). Standard errors are in parentheses. *** *p* < 0.001, ** *p* < 0.01, * *p* < 0.05.

## Data Availability

The original contributions presented in this study are included in the article. Further inquiries can be directed to the corresponding author.

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
