# Peer review of "Consumer Autonomy in Generative AI Services: The Role of Task Difficulty and AI Design Elements in Enhancing Trust, Satisfaction, and Usage Intention"

_behavsci, 2025, doi:10.3390/bs15040534_

Round 1

Reviewer 1 Report

Comments and Suggestions for Authors

Dear author/s,
I have carefully read the article, attempting to evaluate its relevance, execution, and adequacy of the methodology to the objectives.
The objective of the research is relevant, and I find its pertinence well justified. However, I find the literature review to be insufficient, a review that should encompass all the concepts used later in the empirical section. Similarly, I find an inconsistency in the use of the same denominations for concepts in this part. I also believe that hypotheses should be established in this section or, at least, research questions after the justification of those proposed relationships.
For empirical research studies, I find the design of the studies pertinent and good. However the reader would miss a better understanding of the measurements used, their reliability or manipulation check tests. 
While the data have the potential to explain, the analyses run are not well justified or suited for the purpose. No need to carry out three different analyses for the same output, you should choose the one suited for the aims of the study given the nature of the variables used. Therefore, many conclusions from results are not backed up by data.
When comparing groups, it is important to measure the reliability of constructs, invariance between groups and if more than 3, compared by pairs. In some analyses with more groups done the test of F ANOVA only indicates that there are differences between any of these groups, but no showing between which ones are the differences. When comparing the magnitude of the effects in different groups (study 2) you have to conduct a test to check if the difference between these effects is statically significant. 

I may suggest some particular comments on the document:
I find the denomination of the “continued usage intention” concept not accurate. It is redundant to use “continued” as it is not stated in the definition or the item used that it will be continuous, just usage intention or willingness to use. 
Sometimes it is said that complexity influences, and others that it moderates, but the moderation effect is not tested. The same with mediation, it is not statistically tested if it is a complete or partial moderation effect.
I found it not clearly clarified whether autonomy is posed as a design feature or a consumer perception, and it is used afterwards in different ways between studies 1 and 2. Sometimes it is unclear, for instance: “… greatest autonomy is associated with…”
Casual relationships between trust and autonomy should be solidly backed by theory to establish the direction of the relationship.
Despite the shortcomings pointed out, I recognize that the article addresses a relevant topic with important data collection work with potential.

Reviewer 2 Report

Comments and Suggestions for Authors

The article addresses current and important issues concerning considerations on the design of AI services that actively engage consumers in decision-making while maintaining trust and transparency. The analysis carried out confirms the view that designing AI that increases autonomy is not just an additional function, but a necessary basis for ethical and transparent action focused on the development of AI. As AI services integrate with the everyday decision-making process, providing consumers with autonomy will be crucial to building trust and commitment in the responsible use of AI.

The structure of the article is correct, the sample and research tool are correctly selected, although the monetary incentive (approximately USD 1) after completing the study raises ethical doubts. Points that require explanation and correction were also noted:

  1. No directly defined purpose of the article was found. Of course, it is possible to conclude what the purpose is, but it is better to write and define it precisely.
  2. The main purpose of the article should be precisely defined in the abstract and at the end of the introduction. There are two correctly formulated research questions, but in my opinion this is insufficient.
  3. It is also worth answering the question why no research hypothesis was put forward.

Reviewer 3 Report

Comments and Suggestions for Authors

The article is well-written, addresses a highly relevant topic, and is engaging to read. I believe it has significant potential, with only a few aspects that may require further consideration. As you move forward toward publication, I would like to offer some feedback:

  • Readability: The article is generally well-structured and enjoyable to read. However, I have some concerns regarding the organization of the empirical section.
  • Hypotheses: The paper does not offer hypotheses. While it is not mandatory to include hypotheses, it would be helpful to elaborate in your response to the reviewers on the rationale behind this decision.
  • Analysis: The analytical approach could benefit from additional clarification. In both studies, you conduct (1) t-tests, (2) correlation analyses, and (3) regressions. Which of these methods is most critical in addressing your research question? The analysis would be more transparent and interpretable if explicit hypotheses were formulated and tested using a specific method.
  • Analysis Structure: The current organization makes it challenging for readers to follow the analysis. For example, the group-level comparison (line 310) is included under “Manipulation Check,” while correlation and regression analyses are placed under “Results.” This structure may lead readers to question the final conclusions. Clearly stated hypotheses would help mitigate this issue.
  • Conceptual Model: Consider incorporating conceptual models to visually represent the assumed effects and relationships. If you choose not to include such models, please provide a justification for this decision in your reviewer response.
  • Grouping in Analysis: The rationale behind the analytical choices regarding groups is unclear. Why is the correlation analysis conducted across all groups, whereas the regression analysis is performed at the group level? Clarifying this methodological decision would enhance the transparency of your study.
  • Survey Items: Please include the full survey in the appendix. Additionally, when first mentioning the survey (line 248), refer explicitly to the appendix.
  • Item Sources: Ensure that the source of each survey item is clearly communicated.
  • Manipulation Check: Please specify which statistical method was used for the manipulation check.

I wish you all the best as you move forward with the publication process.

Round 2

Reviewer 1 Report

Comments and Suggestions for Authors

Dear Authors,

Thank you for submitting the revised version of your manuscript and for your detailed response to the previous round of reviews.

I am pleased to report that the revisions made are substantial and have significantly improved the manuscript. I appreciate the careful attention you have paid to the comments raised in the first review.

The major issues identified previously have been successfully addressed. In particular, the manuscript's clarity and organization are now much stronger, making the arguments and findings significantly easier to follow. Furthermore, the improvements in the theoretical grounding and methodological justification have substantially strengthened the paper's contribution and rigor. The overall understanding of the study and its implications is greatly enhanced.

As a result of these effective revisions, the manuscript is now a well-structured, clear, and methodologically parsimonious. The concerns raised in my initial review have been satisfactorily resolved.